# Words and Images Matter: Perspectives on Suicide, Mental Health Concerns and Alcohol and Other Drug Use Depiction

**DOI:** 10.3390/healthcare12212120

**Published:** 2024-10-24

**Authors:** Dara L. Sampson, Hannah Cootes, Elizabeth Paton, Jennifer Peprah, Danielle Simmonette, Milena Heinsch, Frances Kay-Lambkin, Jaelea Skehan

**Affiliations:** 1College of Health, Medicine and Wellbeing, University of Newcastle, Callaghan 2308, Australia; hannah.cootes@newcastle.edu.au (H.C.); danielle.simmonette@newcastle.edu.au (D.S.); milena.heinsch@utas.edu.au (M.H.); frances.kaylambkin@newcastle.edu.au (F.K.-L.); jaelea.skehan@health.nsw.gov.au (J.S.); 2Everymind, Newcastle 2300, Australia; elizabeth.paton@health.nsw.gov.au (E.P.); jennifer.peprah@uon.edu.au (J.P.); 3College of Human and Social Futures, University of Newcastle, Callaghan 2308, Australia; 4School of Social Sciences, University of Tasmania and Australia, Hobart 7000, Australia

**Keywords:** mental health, alcohol and other drug (AOD) use, suicide, words, language, communication, images

## Abstract

Background/objectives: The way in which topics like suicide, mental health concerns and alcohol and other drug use are communicated matters. It has the potential to have either a positive or negative impact on people and communities, particularly those with a lived experience of these concerns. This article draws on the findings of a qualitative study designed to explore the experiences and perceptions of stakeholders on the imagery and language used to depict suicide, mental health concerns or alcohol and other drug use. Methods: The focus group method was used as a form of participatory action research to gain an in-depth understanding of the experiences and views of those who use or are impacted by language and imagery about suicide, mental ill-health and AOD use, including those with lived experiences of these topics. Results: A series of 10 focus groups were created in February and March 2022 with media and other professional communicators; people identifying as having a lived experience of suicide, mental ill-health or alcohol and other drug use; mental health and suicide prevention sector professionals; and people from priority populations (*n* = 49). From these focus groups, principles were developed as well as exemplars of helpful and less helpful depictions. Rather than prescriptive or static rules, the participants indicated that safe representations require an ongoing engagement with the principle of “do no harm”. Conclusions: A positive conclusion arose—that words and images have the potential to promote help-seeking, challenge stigma or stereotypes and create change.

## 1. Introduction

How we communicate about suicide, mental health concerns and alcohol and other drug (AOD) use matters, with the potential to have either a positive or negative impact on people in our audiences and communities, including those with a lived or living experience. Some ways of communicating about suicide, for example, have been associated with increased rates of suicide deaths or attempts [1,2,3]. Research shows a particularly strong association between public communication, such as a media report or a social media post, and increased rates of suicide when the method or public location used is detailed or the story focuses on a public figure who has died by or attempted suicide. In combination, including the details of how a prominent person died by suicide, studies showed a 30% increase on average in deaths using the same method [2].

Similarly, the words and images we use to communicate about suicide, mental health concerns or AOD use can perpetuate or contribute to stigma, defined as negative perceptions or beliefs about a person or group based on a particular attribute. For people with a lived or living experience, stigma can be a major barrier to seeking care, treatment adherence and recovery and can contribute to a reduced quality of life, low self-esteem and isolation [4,5,6,7,8,9,10,11]. These negative impacts can result from both news and information media as well as fictional depictions [11,12].

In contrast, there are ways of communicating about these issues that may be positive, protective or preventative. For example, communication about how to cope with distress or adverse circumstances, or stories of people with a lived experience of suicidal thoughts that focus on recovery, may help to reduce suicidal ideation or prevent suicide attempts or deaths [13,14,15]. Similarly, sharing stories of people who live with mental health concerns or who use AOD, with an emphasis on hope and recovery or exploring the experiences of family and friends, can help to decrease stigma and increase community understanding and empathy [11,16,17]. Other stigma-reducing elements may include emphasising the importance of seeking help, accuracy and providing expert comment or advice [7,18,19]. This tells us that communication has an important role to play in the depiction of these significant public health issues.

Internationally, and in Australia, work is already being carried out to support best practise communication about suicide, mental health concerns and AOD use. Guidelines exist, for example, for media reporting [20,21,22,23,24,25], social media [26] and stage and screen [25,27]. These provide broad advice, such as ensuring representation is accurate and balanced, including stories of people with lived and living experience as well as other expert commentary or advice about available support services. Some guidance on language is provided. In Australia, these guidelines are supported with a strong implementation programme (Mindframe) with a team working directly with media, professional communicators and other content producers to ensure guidelines and resources are updated as evidence develops; the programme provides training in best practice guidance and real-time support and advice as incidents occur or issues emerge and embeds the guidelines in organisational policies and standards [28,29,30,31].

While such guidelines and implementation programmes have been shown to be impactful and cost-effective prevention activities [32,33,34], there are nonetheless gaps in the guidance provided. For example, while much of the advice is general in nature and can be applied across media (such as news reports, social media posts or fictional films), there is little to no guidance provided on the specific use of still images, video footage or audio recordings. This is partly owing to the limited evidence on or understanding of the impact these specific elements may have, with most research on media effects in this area being focused on the written components of print-based media [35].

The specific words or language to be used in communicating about suicide, mental health concerns and AOD use is another area where guidance could be deepened. There is a lack of consistency with the language used to communicate about these issues in Australia and a need for greater awareness of its impact. Language has evolved over time as our understanding of the causes and impacts of health issues has changed [36]. However, minimal work has been carried out interrogating the cultural specificity of language around these issues and whether a common language would help to increase understanding of these complex issues and reduce the level of stigma surrounding them. These gaps in the academic literature highlight a need to explore what constitutes the best practise use of images and language and, indeed, whether this is an attainable goal given the complex terrain and changing nature of words and imagery usage. Therefore, the aims of this study were to understand peoples’ experiences and opinions on the use and impacts of imagery and language when communicating about these issues and to potentially fill these gaps in the literature.

In this paper, we report on a subset of qualitative findings from a mixed-method study. This study sought to better understand the experiences, observations and opinions of a broad range of stakeholders on the ways in which language and imagery have developed or reinforced existing stereotypes or other ways of “constructing” or creating discourses about people with lived and living experiences. The research conducted in this study was undertaken as part of a wider project, funded by the Australian National Mental Health Commission, to develop new guidelines and resources focused on language and image use related to mental health and wellbeing, mental health concerns, suicide and AOD use. The findings presented here contribute to the final guidelines alongside the predominantly qualitative results obtained from roundtable discussions, scoping reviews of the literature and current use, a survey on current attitudes, a Delphi consensus study and user testing.

## 2. Materials and Methods

### 2.1. Focus Groups

The focus group method aids in efforts to gain an in-depth understanding of the experiences and views of those who use or are impacted by language and imagery about suicide, mental ill-health and AOD use, including those with lived experiences of these issues. As a form of participatory action research, focus groups empower participants and promote social and cultural change [37,38], while also bridging the (sometimes wide) gap between current research and people’s current experiences [39,40]. This study was underpinned by the theoretical framework of social constructionism, in recognition that how we use language can create or perpetuate power imbalances, while also having the capacity to challenge some of the stigmatising beliefs or stereotypes about people with lived and living experiences. Two conceptual ideas seminal to this theory (critical de-construction and re-construction) were applied to the methodological design in the formulation of the focus group questions and discursive environment created within those focus groups. Different societies and cultures value different things. In this way, social norms are created. What is valued within a society becomes “knowledge” and is therefore constructed, as it is developed and maintained, within a particular culture. Language and imagery support and even define and create how we relate to people with a mental illness or AOD usage issue. Put simply, culturally agreed upon words and images create what is deemed to be “knowledge”. This holds true for linguistic and symbolic signifiers, which can create stigma about a person or their experience [41,42,43]. Language and imagery can construct a particular type of “reality” [44].

The focus groups were designed to de-construct some of these assumed sets of “truths” or “knowledge”; to unpack and question what the basis is for defining something as being “knowledge”. Following this, reconstruction at both the individual and collective levels provided rich, alternate perspectives. We are all influenced by the culture, norms, ideologies and preconceptions in our environment, and therefore, the words and images we use to depict people with a mental health concern or substance use issue will be shaped by our influences, experiences and interactions [45]. This is our “positionality” and influences how we connect to an experience—what we bring and it impacts interactions we might have. In this way, the study methodology was informed by a conscious awareness that each focus group participant, facilitator and researcher brought experiences and pre-conceptions as well as the capacity to create alternative constructions through the process.

For this reason, a constructivist lens was also applied to the discussion of the findings to see how this approach “made sense” of those findings. Words and images construct stories in the public domain that can portray people in a manner that upholds existing power imbalances. This can serve to make the person with the lived experience “different” through the process of “othering” [46]. To date, there is no specific application of the social constructionism theory to the communication experiences of people with lived experiences of suicide, mental health concerns and AOD use in the Australian context. This paper thus extends the existing literature in this area.

Predicated and designed with social constructionism as the conceptual guiding theory, we adopted a strengths-based, appreciative inquiry theoretical framework as a foundation for the focus groups [47,48].

### 2.2. Participants and Recruitment

We used non-probability convenience sampling to identify eligible participants across Australia, drawing on the professional and lived-experience networks of the Everymind team to identify potential participants. Everymind, a national mental health and suicide prevention institute, has extensive networks across multiple sectors, including the Australian media, mental health, suicide prevention and alcohol and other drug sectors, and lived experience organisations. Participants were recruited from within these networks through direct emailing, media alerts and posts on social media. Participants were eligible to participate in the focus groups if they were aged 18 years or older, currently living in Australia and able to participate using the English language.

Participants were requested to self-nominate themselves into a focus group that aligned with their personal or professional experiences. At the time of recruitment, these focus groups included three broad categories: (1) professional communicators (media or sector-based communication professional), (2) people with a professional or personal experience of suicide, mental ill-health or AOD and (3) people identifying as or working with priority populations (young people, men, LGBTIQ+, Culturally and Linguistically Diverse (CALD) populations and Aboriginal and Torres Strait Islander peoples). Given that many participants may have had intersecting identities or personal and professional experiences, participants were able to self-nominate a single or primary focus group of interest and/or multiple focus groups of interest. Participants who self-nominated a single focus group were automatically allocated into this group. Those who self-nominated across multiple focus groups of interest were assigned based on availability and/or registration numbers across groups.

The initial strategy to allocate participants into experience- or population-specific focus groups contended with wide variances of availability, which meant that participants who were unable to attend their allocated focus group on the arranged dates could not easily join another focus group. To this end, an “open” focus group was also created. These factors, as well as concerns related to COVID-19 and extreme weather events across Australian states, New South Wales (NSW) and Queensland (QLD), meant the Aboriginal and Torres Strait Islander focus group was advised by the First Nations facilitator to be cancelled. A specific cultural review of the guidelines and resources was conducted later in the project to ensure Aboriginal and Torres Strait Islander perspectives were included.

Despite these modifications to offset availability issues and external factors, several focus groups went ahead despite not meeting the researchers’ aim for 6–10 participants per group.

In total, 49 participants attended across ten focus groups: (1) media (FG1; *n* = 3), (2) other professional communicators (FG2; *n* = 6), (3) personal or professional experience of suicide (FG3; *n* = 7), (4) personal or professional experience of mental ill-health (FG4; *n* = 7), (5) personal or professional experience of AOD (FG5; *n* = 2), (6) young people (FG6; *n* = 4), (7) men (FG7; *n* = 3), (8) LGBTIQ+ (FG8; *n* = 9), (9) CALD (FG 9; *n* = 4) and (10) open focus group (FG10; *n* = 4). The numbers above are indicative of actual attendance. As described above, some variance in focus group numbers occurred as participants were able to self-nominate a single or multiple focus group category. Issues relating to lower-than-expected attendance will be discussed later as a limitation of this study.

Out of the 49 focus group participants, 45 participants (92%) completed the optional demographic data pre-survey (results shown in Table 1 below).

### 2.3. Setting

Focus groups were held online via Zoom (2022, version 5.9.6, Zoom Video Communications, San Jose, CA, USA) and conducted over a three-week period in February and March 2022. The decision to host the focus groups online rather than in person was influenced by several factors, including extreme weather conditions across NSW and QLD, the COVID-19 pandemic limiting access to public spaces and participants being located across Australia. The groups lasted approximately 90 min and were facilitated by experienced researchers with qualitative and clinical expertise, including one female research professor and four female doctoral-level or higher accredited social workers.

Facilitators were mindful of the need to set up the online space to make participants feel comfortable. In addition to introducing themselves and providing clear expectations for the participants, facilitators took time to acknowledge the valuable experience each person brought to the group and the importance of honouring different perspectives. While not explicitly directed to, participants utilised the video feature, which allowed for people to see the other members of the group and share more comfortably. Participants were asked a set of five questions around the use of words and images used in public representations of suicide, mental health and alcohol and other drug use. These questions were designed to explore participants’ experiences, perceptions, attitudes, opinions and beliefs about safe, inclusive and non-stigmatising public representations of suicide, mental health concerns and alcohol and other drug use.

While facilitators asked these questions consistently across all focus groups, they were mindful of being flexible in the discussions to ensure the priorities of the participants were heard—recognising person-centred, situated knowledge.

### 2.4. Data Analysis

Focus groups were recorded and transcribed verbatim by a professional transcription service. Transcripts were then analysed thematically by two doctoral-level social work and psychology coders and three senior social work and psychology researchers. Braun and Clarke’s [49,50] reflexive thematic analysis approach was applied inductively to privilege participants’ experiences and perspectives. This approach involves six “fluid” phases that aim to conceptualise patterns of shared meaning, which ultimately produce a coherent set of themes [51].

## 3. Results

The research team identified four overarching main themes, which were anchored with quotes to promote participants’ voices: (1) the importance of story and narrative; (2) diversity is needed, but there are challenges; (3) words and images should be context-specific rather than generic; and a (4) sense of balance, community and connection.

### 3.1. The Importance of Story and Narrative

The participants invariably spoke about the power of words and images to reflect and shape public representations of mental ill-health, suicide and AOD, as they reflected on their own specific experiences, understandings and narratives—often emphasising the importance of the broader context of a story. For example, a participant in the CALD focus group highlighted the power of storytelling to their culture: “There’s something so precious and special, so many histories of storytelling, and that’s a really powerful medium to really get messaging through… when community members talk and they use their own language to describe their own experience, it just clicks”. (Participant 1, FG9)

The discussions between participants and facilitators provided insights into how the effective use of images can help illustrate the complexity and nuances of the words in a story. One professional communicator, for example, reflected on using images to depict mental health recovery terminology: “We talk a bit about the recovery journey as opposed to recovery being the end point. So maybe everything’s a journey and I think imagery is part of that, is you can have a happy image and you can be sad at the same time. You can talk about recovery and then you can have a different experience that brings you back to a similar place. It’s just a journey. It’s not about any one end point”. (Participant 1, FG2)

In addition to positive reflections, the participants warned that public portrayals can be misused to construct unhelpful or harmful depictions. For example, mental health recovery can sometimes be depicted as a “cure”: “There’ll also be a lot of celebrity redemption stories… sort of, “I was at my absolute lowest and now look, I’m a multimillionaire”. And you know, “And now I’m cured and I’m amazing”. (Participant 2, FG2)

While imagery was perceived to be an “incredibly powerful” way to “express a story” in the professional communicator focus groups, the participants reflected that they found the “respectful” use of images to be “much harder than language”. In one example, a participant from the media focus group spoke about the “power” of using “juxtaposition” (before and after) images: “We want images that are illustrative, that carry meaning and are attention grabbing. I mean, that’s the business that we’re in. And obviously there’s nothing more attention-grabbing than images which imply action, violence, et cetera… But we also know that they can be stigmatising. So, we’re aware of the issue, but there’s not an easy solution”. (Participant 1, FG1)

Reflecting on this, another media participant warned against prescriptive efforts and emphasised the importance of the wider “story” and narrative. While objectively supporting a more principled approach to the safe use of words and images, an acknowledgement (which permeated the study’s findings) of the nuance lies in the translation: “You can’t turn principles into prescriptive rules very easily. I guess that’s what I’m trying to say. And which is why it always stays slippery and we’re always dealing with ambiguity…”. (Participant 1, FG1)

Across focus groups, the participants noted that historical shifts in language have helped to challenge stigma, shame or negativity surrounding mental health concerns, suicide or AOD usage. In the men’s focus group, the participants also suggested that the use of images could challenge the negative, gendered stereotype of “men looking really sad and angry”. Finally, the power of positive change to construct a different story was succinctly described by one participant: “The old language about speaking about mental health concerns is so dark and negative and dead end, whereas with recovery-oriented language, there are elements of hope and a brighter future… That it’s part of the human condition to have mental health concerns from time to time”. (Participant 1, FG4)

### 3.2. Diversity Is Needed, but There Are Challenges

Themes of diversity and the associated challenges permeated discussions between the focus group participants and facilitators. While it was acknowledged across groups there is no “one-size-fits-all” notion of safe and inclusive public representations, many participants reflected that diverse populations should be able to locate themselves within depictions. Achieving this representation of diversity was, however, described as fraught with complexity. As one media participant described, “I think that’s just an area where I don’t think anyone’s really getting it a hundred percent right all the time”. (Participant 2, FG1)

When discussing the use of images, many focus groups emphasised the importance of being able to “see people who reflect me”. The participants described a perceived lack of diversity within public representations of mental health concerns, suicide and AOD usage. Many stressed the need to reflect the diversity of society more accurately: “I want to see trans people. I want to see disabled people. I want to see people that look different to the Brady Bunch… So, as a trans person, as a disabled person, as a queer person, I want to see those people and people with different relationships portrayed in media, with their consent of course, in a meaningful way and we’re just not seeing that”. (Participant 1, FG8)

Similarly, the participants in the professional communicators focus group described an industry-wide reluctance to use real-life images to depict anything perceived as “sensitive”. Many said they prefer to use graphics, cartoons or icons to accompany messaging about mental health concerns, suicide or AOD use: “We don’t use images, we use icons or graphics, because we struggle with working in government as an independent agency, we don’t want to, I guess, position ourselves with a certain view picture. You know, is the person too young, is the person too old? Are they very obviously affluent, or things like that. So, we tend to use icons, and graphics… because it feels safer”. (Participant 2, FG1)

Some focus groups also described how a sense of “fear” about using the “wrong” language and images about suicide can inhibit honest or open depictions: “But I have a number of new people in my team, and so particularly the topic of suicide is really new to them. And there’s been a real fear about talking about it, not from a stigma perspective, but a fear that if we talk about it, we might cause it”. (Participant 1, FG2)

For one participant, this fear and reticence serves to mask or hide the story, therefore missing the potential for deep conversation and engagement: “I think that language full stop around suicidality is couched in a rhetoric of fear, and so there’s walking around something which actually makes it worse in my experience… It means that there are no actual touch points with the community because no one seems to be able to communicate in an open and honest way or when they do, they’re so afraid of getting it wrong”. (Participant 1, FG7)

Looking ahead to more open, honest and safe uses of words and images, one participant captured the challenges succinctly: “I think you’re going to always struggle with that because what is considered perfectly respectful and non-offensive one day, the very next day can be shifted and become completely offensive to anybody. The rules of language are changing so readily…”. (Participant 3, FG1)

### 3.3. Words and Images Should Be Context-Specific Rather than Generic

The importance of context was a strong theme across focus groups, with participants indicating that the acceptable, credible or appropriate use of words and images cannot be generated through generic templates. Within focus groups, the participants drew out another nuanced point, namely that principles are best employed when tailored to the individual or community. As a result, the participants emphasised that public representation must be highly relevant to the subject and audience context: “You’ve got to tailor each thing to the demographic… if they see someone that they can identify with and go, “Oh shit, that could be me”. I think that’s more powerful than anything else”. (Participant 1, FG5)

Whilst this was an acknowledged challenge, one participant highlighted the creativity and sense of satisfaction this tailoring of messaging can bring: “So having generic language, these kind of standard ways works for some people, but for other people, it just doesn’t. I mean, and that’s always going to be the challenge… That’s the joy of actually understanding what context you’re communicating in”. (Participant 2, FG7)

The participants in focus groups agreed that the best method to contextualise words and images is to work directly with impacted populations. Taking the perspective of the person most impacted by the language usage creates empathy and can generate goodwill, particularly in highly sensitive topics or areas of personal discomfort: “So, the first point of contact is the person that you’re talking to or the person that you’re talking about, and how do they want to be described and what language do they want to be used in the context of the story that’s about them”. (Participant 1, FG1)

While generic representations were identified as a barrier to safe and inclusive representation, several participants noted that including diverse living and lived experiences can be a complex and challenging task: “…language can unintentionally or intentionally create a sense of safety or lack of safety… the way someone talks will tell me if it’s a safe space and sometimes they might not realise that, they’re creating safety or they’re pushing me away”. (Participant 2, FG8)

Additionally, several focus groups indicated a preference for public representations to move away from deterministic, clinical labels and instead have a broader focus on health and wellbeing: “I think leading with the medicalised language can feel pathologizing… There are so many ways that we can use language to discuss mood, mental health behaviour and open up those conversations”. (Participant 1, FG9)

Across focus groups, the participants suggested the inclusion of terminology outside of clinical diagnoses and symptoms to include a broader focus on health and wellbeing. The participants discussed what they saw as an overreliance on the medicalisation of certain issues, with a perceived emphasis on “deficit-focused” and “negative” representations, rather than ones that are “strength-based” or “holistic- and wellbeing-focused”. For example, some participants took issue with what they saw as the overuse of the term “shocking statistics.” In other groups, the participants reflected that there is a need to avoid certain, overused images in the future: “Can I just add that I just want to say that it’s not just the generic picture as well where you’ve got the hooded person where they’re crying and they’re just in darkness, just sitting, like yeah. I just don’t want to see that as well because that’s just overdone”. (Participant 3, FG8)

The participants also identified the need for increased awareness about the cultural context, expectations and sensitives of specific communities and populations. For example, discussions between the facilitator and participants in the CALD focus group highlighted that even when “diagnostic labels” are used appropriately, from a clinical perspective, they might not necessarily reflect cultural attitudes: “As a person from a different culture talking about some things very openly, for example, the White culture or mainstream culture in Australia, talks about could be considered quite rude”. (Participant 2, FG9)

Communication professionals emphasised that “the purpose” of public representations should always be based on principles of thoughtful, sensitive and constructive communication that “does no harm”. The participants noted that this is about more than “just words”—it is about the substance of the communication process and whether it is constructive. This was said to apply to generic messages intended to promote help-seeking. For participants in the LGBTIQ+ focus group, there were many discussions about the generic help-seeking idea attached to LGBTIQ+ communication being disengaging: “…even the sentence they say before of “if this has triggered anything for you, then we’ll put the lines on the bottom of the screen”. It all sounds like every show I watch… And I think, for me, I don’t even know that I take it in and I certainly don’t feel inspired to call them”. (Participant 4, FG8)

Professional communicators also took issue with generic content warnings and questioned whether they led to unintended negative consequences: “Putting warnings up on stories all the time, excessive numbers of warnings on stories, is a bad idea and it is both ineffective and potentially harmful. And so, the bar that we have traditionally applied to cause harm is relatively high, whereas the consensus was… I wouldn’t say it’s a consensus because I know it’s not a consensus, but a very current view about triggering sets a very, very low bar on triggering”. (Participant 1, FG1)

### 3.4. Sense of Balance, Community and Connection

Across focus groups, the participants reflected that mental health concerns, suicide and AOD usage are unique and deeply personal experiences. Several participants highlighted that people with lived or living experiences of these issues can, and do, live meaningful, satisfying, connected, and hopeful lives—yet this notion is often missing from public representations. The participants suggested that a more balanced “picture” and “messaging” is needed. The benefit of instances where more balanced and perceived hopeful depictions occur was described by participants across all focus groups: “…it’s that sense of community and connection. How do we portray connection instead of isolation? So we don’t need to reinforce the stereotype, we need to give you a different image to replace the one that you’ve got in your head”. (Participant 2, FG2)

The participants also noted that effective public representations of mental health concerns, suicide and AOD can help to challenge the cultural stigma and expectations within communities. For example, some participants reflected that a cultural pressure to remain quiet about mental health concerns exists due to a perceived fear of ostracism for “speaking up”. This can stultify expression and help-seeking and impact self-perception. The participants in the CALD focus group were positive about the power of public representations to help in addressing historical and cultural pressures: “So, a picture of a few people, or a group of people supporting a person who’s going through a mental illness or suicide could be good messaging to get across both ways. For the community to feel like we have something to do to support people who are in need”. (Participant 2, FG9)

There was a discussion about the importance of balancing individualised representations with those that are community-focused in the CALD focus group: “Almost without exception our clients talk about their community… It’s with the encouragement of community members, whether it’s immediate family or wider social network, that’s where the messaging and the support. I love the idea of community-focused mental health messaging”. (Participant 1, FG9)

Without question, the participants with lived experiences stressed that any suggestion of blame, violence, unpredictability, incompetence or social exclusion should be avoided. Finally, the public communicators indicated a need to stop incorrectly using the term “mental health” when the intent is actually describing a “mental illness”. However, they also acknowledged the nuance in balancing accuracy with simple, accessible language for people who may have lower levels of health literacy: “…we talked a lot about being the translators so that we could find the balance between translating highly clinical or evidence-based information or sector language into language that everybody could understand, while at the same time trying to shift language”. (Participant 1, FG2)

## 4. Discussion

We explored the experiences of 49 people through 10 focus groups. The participants were asked to indicate a preference for inclusion as a professional communicator, as a person with professional or personal experience of suicide, mental health concerns or AOD use, or as a person identifying as or working with a particular priority population. It was exceptional for an individual, despite nominating a “category”, to not have a multiplicity of “hats”. In conversations, participants revealed personal stories either about themselves or people with whom they were close. In terms of participant positionality, this embedded professional experiences within personal narratives, rendering added depth to the perspective from which they discussed their expertise. Furthermore, through the discursive style of the focus groups, change was brought about as much by the process as the outcome, as the participants engaged with the ideas of others. In this way, language was the conduit for the participants to explore each other’s words and image preferences. This was a parallel exploration of the constructions we brought to the process and the ways in which these constructions could be shaped by others within the group. Ideas were de-constructed and re-constructed through group process [41,43,52].

The importance of story and narrative was a clear finding. Across focus groups, it was emphasised that specific words and images exist within broader historical, cultural and social conditions. Meanings were said to be contingent on the worldview of the audience, and representations can therefore be interpreted in multiple ways, as is consistent with constructivist theories which challenge the notion of an absolute “reality” which sits outside of the reality that is created socially [41,45]. The participants articulated how stories shape their culture and experiences and, while the participants agreed there are still negative portrayals of people experiencing suicidal behaviour or bereavement, or mental health or substance use concerns, there was also optimism. The participants uniformly demonstrated awareness of the power of language and positive imagery to re-imagine stories, much in the way described by Shotter [52] and Haslanger [45]. Reality and social norms can be influenced by the power of showing different stories and representations. This speaks to the circularity of change. As we are influenced by stories constructed within any culture, we have the power to influence culture through our conscious avoidance of some words and images and our privileging of others. Through this role modelling, the experiences and choices of others can be changed to create a societal shift.

Overall, the findings indicate that the participants see the wider story and narrative as inseparable from the specific word or image choices. The participants emphasised that many stories and narratives are missing, or under-represented, in the public sphere, particularly culturally appropriate and diverse narratives reflecting the diversity of Australia’s population. While vast improvements were achieved over the past decade, more meaningful engagement with narratives about suicide was called for, with the participants noting that it is often spoken of obliquely or avoided [53]. For public communicators, however, it can be a challenge to balance safe and inclusive storytelling with the specific editorial purpose. It was suggested that the principle of “do no harm” be foregrounded for all media professionals in their storytelling. It was further agreed that words and images have the power to help change people’s stories and promote more hopeful, recovery-oriented storytelling. This positions words and images as conduits of change; namely, it is a way to challenge traditional ways in which people with a mental illness or AOD usage issue have been portrayed by those who have held the power of communication, policy making and government to do so [54]. The power created when a person or group of people is given a “voice” was important, and this study provided that voice to people in priority groups, who are often not consulted in decisions about their representation.

Diversity is needed, but there are challenges. Whilst it was acknowledged across groups that there is no “one-size-fits-all” notion of safe and inclusive public representations, many participants reflected that diverse populations should be able to locate themselves within depictions. The participants indicated that there is not enough diversity in who is currently represented in images or story contexts, with priority populations decrying a dearth of representations with which they can relate. When dominant groups determine how priority populations are portrayed, this can further serve “other” people in those groups and emphasise difference over commonality [46]. Public communicators indicated that choosing diverse images in a meaningful way is a noted challenge, and there is an industry-wide reluctance to use real-life images due to fear of unintentionally misrepresenting an individual or reinforcing stigma. Instead, cartoons and graphics are often preferred. This study provides a starting point in this conversation. Language can shape perceptions to form a different “reality” [44]. As language and imagery change, so does our commonly accepted usage of both.

Context is important. One particular focus group discussion (a session with men as the priority population) saw a debate emerge about the nuanced line between being perceived to condone what the listener perceives as inappropriate terminology with engagement in a therapeutic relationship; in this example is a young male who was reticent to seek help. This demonstrates a practise principle of starting from the perspective of the person most impacted. Put simply, for this study, it involved valuing the right of an individual to be the decision maker as to what language and images they preferred. Through our application of appreciative inquiry [47] as a methodological frame, and understanding our own positionality, the experiences and perspectives of those most impacted were given privilege. Thus, this became a key finding and a guiding principle from this study, namely contextualising language and visual images to ensure appropriateness for the intended audience. Beginning with the person and checking their preference is fundamental. This not only denotes respectful practise but also serves to empower the individual by giving them agency over their depiction. This finding presents an explicit challenge for professional and public communicators whose work engages large, and largely unknown, numbers of people with varied and even contradictory language preferences.

Considering the aforementioned challenges, the participants indicated that the acceptable, credible or appropriate use of words and images cannot be a generic consideration. Instead, public representation of mental health concerns, suicide and AOD usage needs to be highly responsive to the subject and audience context. The participants indicated that word and image use should be informed by meaningful engagement with those impacted by the story rather than assumptions or stock images. Additionally, several focus groups indicated a preference for public representations to move away from deterministic, clinical labels and instead have a broader focus on health and wellbeing. Accepted “truths” and “knowledge” were de-constructed, and a challenge was brought to medical models which can construct people as being the sum of their diagnosis.

Closing with a return to a strength-based approach [48] that acknowledges, builds and extends upon what is already working well within a community, striving for a sense of balance, community and connection is important as a strategy to combat stereotyping. Across focus groups, the participants reflected that mental health concerns, suicide and AOD use are unique and deeply personal experiences and should be approached as such. Several participants highlighted that people with lived and living experiences of these issues can (and do) live meaningful, satisfying, connected and hopeful lives—yet this notion is often missing from public representations. The participants suggested that more balanced, strength-based and holistic public representations are needed as mental health concerns, suicide and AOD use can often be depicted as homogenous and negative experiences permeated by darkness and isolation. With no intention of minimising the (sometimes) debilitating impacts, it was noted that safe, inclusive and non-stigmatising representations should aim to balance existing depictions with more hopeful choices. This includes the use of colour (e.g., light rather than dark), images chosen (e.g., people who are supported and part of a wider community rather than alone with their heads in their hands) and stories told (e.g., recovery narratives and those outside of stereotypes).

### Limitations and Implications for Future Research

The findings of this study provide important insights into participants’ complex experiences of, and views on, how to best use words and images to sensitively portray people with a mental health or AOD issue or experiences of suicide. The non-probability convenience sampling method used for participant recruitment to this study has limitations in terms of its generalisability across the whole Australian population. Efforts were made, however, to broaden the findings’ generalisability beyond a small homogenous sample. This included the targeted recruitment of people with a lived experience of suicide, mental ill-health and AOD use (via multiple Australian lived experience organisations and networks) as well as media (via a nationwide media database as well as a national media advisory group) and other professional communicators from the mental health and suicide prevention sectors (via key sector organisations and member networks) who are the primary beneficiaries of the resulting communication guidance. It also included efforts to recruit people from priority population groups (young people, men, LGBTIQ+, CALD and Aboriginal and Torres Strait Islander people) that are each disproportionately represented in suicide, mental ill-health and AOD use data. Alterations to the focus group method, running as mixed rather than experience-/identity-specific groups, may lead to an increased sample in the future. Although there was a commitment to garnering widespread perspectives from identified priority populations, the absence of a deeper understanding of First Nations peoples’ perspectives is acknowledged. This is a clear area for future research.

In conjunction with a broad evidence base and other scoping and consultation work, the findings from these focus groups have been used to inform the development of new guidelines. These aim to aid in the selection of appropriate language and images for use in public communication on suicide, mental health concerns and AOD use. The findings have also informed the development of an image database for communications professionals to illustrate their public communication with non-stigmatising and inclusive imagery. It is important here to also acknowledge the changing landscape in language and imagery preference in describing suicide, mental health concerns and AOD use. This is rapidly moving terrain. What might be considered “best practice” now may be anachronistic in five years’ time. For this reason, a decision was made to develop a set of guiding principles as a key study deliverable. Subsequent to the analysis of the findings, these principles were used to create exemplars of “problematic” and “preferred” language usage. Imagery usage, whilst not unanimous, created less contention. Language was a more contestable environment. This indicates a requirement to review these guidelines regularly. Evaluating these guidelines’ impact on image and language use in Australia presents further research opportunities.

## 5. Conclusions

This collaborative, participatory research approach enabled the collective exploration and sense making of safe, inclusive and non-stigmatising public representations of mental health concerns, suicide and AOD use. This study sought to understand what Australian stakeholders (particularly people with lived experiences) believe are the key issues and priorities to be addressed in the development of words and images guidelines. A thematic analysis of the findings helped to establish how participants want their lived and living experiences depicted in words and images and gave professional communicators and sector professionals a place to share their learnings. Rather than prescriptive or static rules, the participants indicated that safe representations require an ongoing engagement with the principle of “do no harm”. While it was noted that this is a complex and challenging task, the participants shared broad and important insights into what this principled approach might look like. These findings have indicated that the use of stigmatising, isolating and discriminatory words and images to depict suicide, mental health concerns and AOD use is pervasive and has the potential to cause harm. There is a need for representations of these issues that accurately portray a broad range of lived and living experiences.

These findings, in the context of the wider project, have the potential to contribute to safer, more inclusive representations. Indeed, while there was a complex interplay of different perspectives within and across the focus groups, there was a shared perspective—that words and images have the potential to promote help-seeking, challenge stigma or stereotypes and initiate changes in understanding. This is a profoundly important paradigm for understanding the perspectives and challenges for people with lived and living experiences, with the potential to change the way we culturally conceptualise and portray their stories.

## Figures and Tables

**Table 1 healthcare-12-02120-t001:** Demographics of participants.

Demographic Category	Results
Gender	Woman or female (*n* = 34, 76%), man or male (*n* = 7), non-binary (*n* = 2), trans man (*n* = 1), prefer not to say (*n* = 1)
Country of birth	Australia (*n* = 37, 82%), United Kingdom (*n* = 1), Saudi Arabia (*n* = 1), East Africa (*n* = 1), Indonesia (*n* = 1), prefer not to say (*n* = 1)
Language spoken at home	English (*n* = 44, 98%); Arabic (*n* = 1)
Cultural background or ethnicity (participants able to list more than one)	Australian (*n* = 37, 82%), English (*n* = 5), Scottish (*n* = 4), German (*n* = 3), Irish (*n* = 1), Chinese (*n* = 1), French, (*n* = 1), Palestinian (*n* = 1), Ukrainian (*n* = 1), Polish (*n* = 1), Jewish (*n* = 1), Indonesian (*n* = 1)
Age	Mean age 44.46 (youngest 23 years old; oldest 70 years old)
Sexuality	Straight (heterosexual) (*n* = 38, 84%), queer (*n* = 2), gay (*n* = 2), lesbian (*n* = 2), bisexual (*n* = 1), pansexual (*n* = 1), prefer not to say (*n* = 1)

## Data Availability

The data are available upon reasonable request to D.L.S. (dara.sampson@newcastle.edu.au).

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
