# Peer review of "Words and Images Matter: Perspectives on Suicide, Mental Health Concerns and Alcohol and Other Drug Use Depiction"

_healthcare, 2024, doi:10.3390/healthcare12212120_

Round 1

Reviewer 1 Report

Comments and Suggestions for Authors Dear authors,   First of all, I want to congratulate you for your research and also for this paper. It very relevant for professionals working in the prevention, harm reduction and communication fields. The text is clear and well written, however I recommend you to do minor changes to make you paper more readable and the data better contextualized. I’m sending below few comments:   - On the section 2.2 you describe the participants and recruitment. However, I recommend you to add a table describing the participants. It is also recommended to identify the number of focus group fper group (this info is missing). You can keep them aggregated by their profile, but it would be easier for the read or to have this table to better contextualize the results. If you collect them, it would be also relevant to add sociodemographic data (e.g., age range, gender). For example, you can create a table with a line for each group and 3 columns (1- identifying the profile, 2- the number of the group (e.g. FG1) and respective number os participants per FG, 3 - sociodemographic information of the participants).  - In the results section, you present several extracts of the focus groups. However, the data would be more readable and rigorous if the extracts are clearly identified, (e.g., participant 2_FG1).   Warm wishes, The reviewer  Comments on the Quality of English Language

- The text is clear and well written. However, I recommend a proofread to the text to correct minor typos or to reformulate sentences (e.g., the first sentence of the abstract; “on this topic” line 131-132 its not needed; the “s” on the line 162). I suggest you proofread the document.

Author Response

On the section 2.2 you describe the participants and recruitment. However, I recommend you to add a table describing the participants. It is also recommended to identify the number of focus group fper group (this info is missing). You can keep them aggregated by their profile, but it would be easier for the read or to have this table to better contextualize the results.

We thank the reviewer for their thoughtful review and have amended 2.2 accordingly to include a table, demographics and number of focus groups.

If you collect them, it would be also relevant to add sociodemographic data (e.g., age range, gender). For example, you can create a table with a line for each group and 3 columns (1- identifying the profile, 2- the number of the group (e.g. FG1) and respective number os participants per FG, 3 - sociodemographic information of the participants).  - In the results section, you present several extracts of the focus groups. However, the data would be more readable and rigorous if the extracts are clearly identified, (e.g., participant 2_FG1).

Demographics added as suggested. Participants now identified throughout by focus group of origin.

Reviewer 2 Report

Comments and Suggestions for Authors

This study aims to report the findings of a qualitative study designed to explore  the experiences and perceptions of stakeholders on imagery and language used to depict suicide, mental health concerns or alcohol and other drug use.  The article addresses a topic that is both original and worthy of further research in terms of content. It is also aligned with the scope of the journal. The conceptual framework of the research has been well prepared and the relevant literature has been well analysed. However, there are some significant issues with the research that require correction by the authors. I will list these below: 

The abstract should be revised to include a detailed account of the methodology, the study group, and the timing of the study. The abstract currently lacks sufficient information on these aspects. 

The methodology employed in the formation of the study group and the distribution of the study group appears to be one of the most contentious aspects of the study. In the concluding paragraph of page 3, the study group is delineated as follows: "Within the priority populations category was a subset of six groups; young people (n=4); Culturally and Linguistically Diverse (CALD) populations (n=4); LGBTIQ+ (n=9); men (n=3) and Aboriginal and Torres Strait Islander peoples. "  It is essential to provide a detailed account of the methodology employed in the formation of the study group and the rationale behind the number of participants selected for each group. For instance, the rationale behind the higher number of participants in the LGBTIQ+ group compared to other groups should be elucidated in a comprehensive manner. 

The authors used non-probability convenience sampling to identify eligible participants across Australia. The non-probability convenience sampling method is an inadequate sampling method with regard to the generalisability of the results of the research, and this should be emphasised in the limitations section of the research. 

The researchers utilised the Everymind organisation's network to identify potential participants. Please provide a brief overview of this organisation, as not all international readers may be aware of its existence. 

While the language employed in the research is consistent with that used in academic writing, there are instances where the use of lengthy sentences impedes readability and comprehension of the study. For instance, the first paragraph of the conclusion section on page 11 is as follows: "The findings from these focus groups, in conjunction with a broad evidence base and other scoping and consultation work, have been used to inform the development of guidelines to aid in the selection of appropriate language and images for use in public commu nication on suicide, mental health concerns, and AOD as well as inform the development of an image database for communications professionals to utilise to illustrate their public communication with non-stigmatising and inclusive imagery. "  It would be beneficial to streamline sentences that are unnecessarily lengthy, as this would enhance comprehension of the text.

Author Response

The abstract should be revised to include a detailed account of the methodology, the study group, and the timing of the study. The abstract currently lacks sufficient information on these aspects. 

We thank the reviewer for their kind suggestions and provide the following response to suggested improvements. The abstract has been amended accordingly.

The methodology employed in the formation of the study group and the distribution of the study group appears to be one of the most contentious aspects of the study. In the concluding paragraph of page 3, the study group is delineated as follows: "Within the priority populations category was a subset of six groups; young people (n=4); Culturally and Linguistically Diverse (CALD) populations (n=4); LGBTIQ+ (n=9); men (n=3) and Aboriginal and Torres Strait Islander peoples. "  It is essential to provide a detailed account of the methodology employed in the formation of the study group and the rationale behind the number of participants selected for each group. For instance, the rationale behind the higher number of participants in the LGBTIQ+ group compared to other groups should be elucidated in a comprehensive manner. 

Section 2.2 has been amended significantly (highlighted text) to address the concerns of the reviewer.

The authors used non-probability convenience sampling to identify eligible participants across Australia. The non-probability convenience sampling method is an inadequate sampling method with regard to the generalisability of the results of the research, and this should be emphasised in the limitations section of the research. 

Section 4.1 has been re-written to address the limitations of this type of sampling.

The researchers utilised the Everymind organisation's network to identify potential participants. Please provide a brief overview of this organisation, as not all international readers may be aware of its existence. 

Addressed 2.2 line 155.

While the language employed in the research is consistent with that used in academic writing, there are instances where the use of lengthy sentences impedes readability and comprehension of the study. For instance, the first paragraph of the conclusion section on page 11 is as follows: "The findings from these focus groups, in conjunction with a broad evidence base and other scoping and consultation work, have been used to inform the development of guidelines to aid in the selection of appropriate language and images for use in public communication on suicide, mental health concerns, and AOD as well as inform the development of an image database for communications professionals to utilise to illustrate their public communication with non-stigmatising and inclusive imagery. "  It would be beneficial to streamline sentences that are unnecessarily lengthy, as this would enhance comprehension of the text.

Line 575 has been re-worded for clarity and brevity and a general check has been conducted throughout the paper.

Reviewer 3 Report

Comments and Suggestions for Authors

Dear Authors;

It is an article with plenty of information on the subject, nevertheless, the following recommendations are made to improve the quality of the article:

. In order to have a clear and complete idea of the study, it would be appropriate to include the study sample in the abstract.

. The aim of the study is unclear. Please formulate it in such a way that it is easy to find.

. Line 151. When acronyms or initials are mentioned to me for the first time, it is appropriate to include their meaning (NSW and QLD).

. Revise spelling in lines 162

. The information throughout the document becomes monotonous and not very dynamic as it does not include any table or image about this research, which makes it a heavy dialogue and not very motivating to read. It is suggested that an image or table be introduced to make the document dynamic and motivate discussion.

. Lack of support from other research or scientific papers, paragraph (425-441).

Best regards

Author Response

In order to have a clear and complete idea of the study, it would be appropriate to include the study sample in the abstract.

The authors appreciate the suggestions made by this reviewer. We have now included the sample size in the abstract.

The aim of the study is unclear. Please formulate it in such a way that it is easy to find.

This has been added to the Introduction at line 91.

Line 151. When acronyms or initials are mentioned to me for the first time, it is appropriate to include their meaning (NSW and QLD).

This amendment has been made and is now found at line 180.

 Revise spelling in lines 162

Corrected.

The information throughout the document becomes monotonous and not very dynamic as it does not include any table or image about this research, which makes it a heavy dialogue and not very motivating to read. It is suggested that an image or table be introduced to make the document dynamic and motivate discussion.

A table has now been included at this reviewer's suggestion.

Lack of support from other research or scientific papers, paragraph (425-441).

This is noted and, in response, an additional six academic references have been included and are highlighted in bibliography.

Round 2

Reviewer 2 Report

Comments and Suggestions for Authors

It is evident that the majority of the recommendations I proposed to the researchers regarding the study have been incorporated.

The authors re-write the abstract section in a more detailed way containing the study group. 

I am glad to see a more comprehensive account of the distribution of the study group, particularly within the methodology section. Furthermore, the presentation of the detailed characteristics of the participants in tabular form was both elucidating and suitable. 

The remaining sections of the manuscript have been revised and edited in a satisfactory manner. Therefore, it is recommended that the manuscript be published.